# Foliar Fungal Endophytes in a Tree Diversity Experiment Are Driven by the Identity but Not the Diversity of Tree Species

**DOI:** 10.3390/life11101081

**Published:** 2021-10-13

**Authors:** Stephan Kambach, Christopher Sadlowski, Derek Peršoh, Marco Alexandre Guerreiro, Harald Auge, Oliver Röhl, Helge Bruelheide

**Affiliations:** 1Institute of Biology/Geobotany and Botanical Garden, Martin Luther University Halle-Wittenberg, Am Kirchtor 1, 06108 Halle, Germany; helge.bruelheide@botanik.uni-halle.de; 2Faculty of Biology and Biotechnology, Ruhr University of Bochum, Universitätsstraße 150, 44801 Bochum, Germany; christopher.sadlowski@rub.de (C.S.); derek.persoh@rub.de (D.P.); marco.guerreiro@rub.de (M.A.G.); oliver.roehl@rub.de (O.R.); 3Department of Community Ecology, Helmholtz Centre for Environmental Research—UFZ, Theodor-Lieser-Str. 4, 06120 Halle, Germany; harald.auge@ufz.de; 4German Centre for Integrative Biodiversity Research (iDiv) Halle-Jena-Leipzig, Puschstraße 4, 04103 Leipzig, Germany

**Keywords:** biodiversity–ecosystem functioning, cryptic, *Fagus sylvatica*, leaf fungi, *Quercus petraea*, *Picea abies*, Shannon diversity, species richness

## Abstract

Symbiotic foliar fungal endophytes can have beneficial effects on host trees and might alleviate climate-induced stressors. Whether and how the community of foliar endophytes is dependent on the tree neighborhood is still under debate with contradicting results from different tree diversity experiments. Here, we present our finding regarding the effect of the tree neighborhood from the temperate, densely planted and 12-years-old Kreinitz tree diversity experiment. We used linear models, redundancy analysis, Procrustes analysis and Holm-corrected multiple *t*-tests to quantify the effects of the plot-level tree neighborhood on the diversity and composition of foliar fungal endophytes in *Fagus sylvatica*, *Quercus petraea* and *Picea abies*. Against our expectations, we did not find an effect of tree diversity on endophyte diversity. Endophyte composition, however, was driven by the identity of the host species. Thirteen endophytes where overabundant in tree species mixtures, which might indicate frequent spillover or positive interactions between foliar endophytes. The independence of the diversity of endophytes from the diversity of tree species might be attributed to the small plot size and the high density of tree individuals. However, the mechanistic causes for these cryptic relationships still remain to be uncovered.

## 1. Introduction

In forest ecosystems, the biodiversity and functioning across all trophic levels, from the decomposing community in the soil to the predator community in the sky, are driven by the diversity and composition of the tree community [1]. Due to their hidden nature, the diversity and functioning of cryptic organisms (e.g., bacteria and fungi within plant roots and leaves) are less well studied than the more conspicuous organism groups. However, the diversity and interactions between cryptic organisms can shape the functioning of ecosystems [2] and might even alleviate the effects of climate change on plants [3,4]. Cryptic organisms can significantly drive the functioning of a forest ecosystem via the release of bioactive compounds that alter nutrient uptake, tree growth, resistance to herbivores and pathogens, stress tolerance and soil nutrient cycling [5,6].

Foliar endophytic fungi (FEF) reside within leaf tissue and exhibit beneficial effects on the host plant. In trees, FEF can trigger effects on leaf productivity [7], plant growth [8], draught tolerance [9], resistance against pathogens and pests [8,10,11,12,13] and, furthermore, moderate the nutrient cycle via the quality and decomposition rates of leaf litter [8,14]. In summary, FEF could play a significant role in shaping the observed positive effects of tree diversity on the functioning of a forest ecosystem [15] and might provide a mechanism to alleviate climate-change related stressors. To predict the effects of tree diversity on forest diversity and functioning, its effect on the FEF community must also be understood.

The species identity, the genotype and the age of the host tree can all affect the diversity of the FEF community [8]. Beside those host-specific effects, the diversity of FEFs was found to differ between sampling sites and to be correlated with temperature and precipitation regimes, indicating that local abiotic conditions might filter for certain life-history traits in the FEF community [8]. Whether and how the diversity of FEF is also shaped by local biotic conditions, namely the composition and diversity of the local tree neighborhood, is still a key issue in forest biodiversity–ecosystem functioning research [16]. The study of tree diversity experiments should provide a mechanistic understanding of the effects of the diversity and composition of the neighboring tree community on the community of FEF. However, the empirical findings from tree diversity experiments are still inconsistent. In a five-year-old and densely planted experiment in Canada, Laforest-Lapointe et al. [15] found a positive effect of tree diversity on the diversity of bacterial foliar endophytes. In a three-year-old and more sparsely planted experiment in the US, Griffin et al. [17] found the opposite relationship. In the largest tree diversity experiment in China, Saadani et al. [18] found that foliar fungi in general (including endophytic and pathogenic species) were not affected by the diversity of the tree neighborhood. Tentatively, one might speculate that the effects of tree diversity on FEF might increase with the age and the density of the planted trees. Still, additional observations from different tree diversity experiments are urgently needed to put the published findings into perspective.

In this study, we sampled the FEF community in a 12-years-old and densely planted tree diversity experiment near Kreinitz (Germany) to test the effects of the tree neighborhood on the diversity and composition of the FEF in beech (*Fagus sylvatica* L.), oak (*Quercus petraea* (Matt.) Liebl.) and spruce (*Picea abies* (L.) H. Karst.). Previous results from this experiment revealed that the host identity and the diversity of the tree neighborhood drive the infestation and richness of pathogenic fungi in tree leaves, as assessed by microscopy [19].

In this study, we went from the microscopic to the molecular scale by using an ITS marker to identify the FEF community in the Kreinitz tree diversity experiment. We hypothesized that beech, oak and spruce samples harbor a different diversity and composition of FEF. We expected that the tree neighborhood drives the composition and the diversity of FEF. We hypothesized that the similarity in the sampled FEF declines with the distance between sampled plots. Finally, we tested if there are certain FEF that are restricted to monospecific or mixed tree communities. With these analyses, we provide an extensive assessment on the effects of the tree neighborhood on the community of FFE in a 12-years-old and densely planted tree diversity experiment.

## 2. Materials and Methods

### 2.1. Leaf Sampling

The Kreinitz tree diversity experiment was established in 2005 by the Helmholtz Centre for Environmental Research—UFZ near the village of Kreinitz, Germany (51.385616 N, 13.261539 E) to test for the effects of the functional diversity of the planted tree communities on ecosystem functioning (treedivnet.ugent.be/ExpKreinitz.html). It consists of two replicated blocks, each containing 49 plots of 5 m × 5 m with all monocultures and all possible mixtures of two, three, five and six species of the following native tree species: European beech (*Fagus sylvatica* L.), European ash (*Fraxinus excelsior* L.), Norway spruce (*Picea abies* (L.) H. Karst.), Scots pine (*Pinus sylvestris* L.), Sessile Oak (*Quercus petraea* (Matt.) Liebl.) and Small-leaved Lime (*Tilia cordata* Mill.). Within each plot, tree species were randomly planted in equal proportions of a total of 36 individual trees, in 6 rows at a distance of 1 m within and 0.8 m distance between rows.

In July and August of 2017, we collected leaf samples from two deciduous (beech and oak) and one evergreen tree species (spruce). Within each plot, we randomly selected two individuals of each focal species (avoiding direct neighbors) from which we collected four leaves (beech and oak) or four twigs (spruce), each from one of the main compass directions. Samples thus consisted of eight leaves of beech or oak and eight twigs of spruce. All samples were collected at breast height or the nearest available height, clipped in the morning and kept in an ice chest until further processing.

We sterilized the leaf surface of each sample according to the following protocol (modified from [20] and [14]): 1 min of washing in 500 mL ddH_2_O, 2 min in 70% ethanol, 5 min in 1.4% sodium hypochlorite, followed by three additional washing steps of 1 min in 250 mL ddH_2_O. From each sample of beech and oak, we clipped 32 circular discs with a steel cork driller. From each sample of spruce, we clipped 32 needles. These leaf materials were dried and stored on silica gel until further processing.

### 2.2. DNA Extraction and Sequencing

The dried leaf material was disrupted and homogenized in a FastPrep®-24 Instrument (MP Biomedicals, Eschwege, Germany) at 6 m s−1 for 60 s after adding a bead mixture (0.03 g of Ø 0.1–0.25 mm, 0.06 g of Ø 0.25–0.5 mm and 5 or 6 glass beads Ø 1.25–1.55 mm) to the samples. The DNA was subsequently extracted using a gDNA Plant Kit (Invitrogen™, ChargeSwitch®, Thermo Fisher, Munich, Germany) according to the manufacturer’s instructions, with all volumes scaled down to 10%.

The fungal barcoding region (i.e., ITS rRNA gene region) [21,22] was amplified and sequenced to assess the composition of the FEF communities. The amplicon libraries were prepared and sequenced as detailed by Gossner et al. [23]. Briefly, the amplicon libraries comprised two amplification steps. The first PCR was performed with the fungal specific primer combination ITS1F [24] and ITS4 [25], modified to include TAG sequences and part of the sequencing primers (Appendix A). The second PCR was performed with primers that included the sequencing adapters for Illumina sequencing. The final PCR products were pooled equimolarly and purified with CleanPCR® Nucleic acid Clean up (CleanNA, GC biotech B.V., Waddinxveen, Netherlands) according to the manufacturer’s instructions. Sequencing was performed by the sequencing service of the Faculty of Biology at LMU Munich, using an Illumina MiSeq® sequencer (Illumina, Inc., San Diego, CA, USA) with 2 × 250 bp paired end sequencing (MiSeq Reagent Kit v3 Chemistry, Illumina, Inc., San Diego, CA, USA).

### 2.3. Sequence Processing

The obtained raw Illumina reads were processed according to Röhl et al. [26] by using the Qiime 1.9.1 software [27]. Briefly, the TAG sequences were extracted, and the reads were quality filtered with Phred >19. The samples were demultiplexed based on their indices and TAGs (Appendix A). The preprocessed reads were deposited in the European Nucleotide Archive (www.ebi.ac.uk/ena/browser/view/PRJEB45840). The processed forward reads, comprising the ITS1 region, were further clustered into Operational Taxonomic Units (OTUs) by using CD-HIT-OTU for Illumina reads version 0.0.1 [28] (http://weizhongli-lab.org/cd-hit-otu) with a 97% similarity threshold. The representative sequence for each OTU was assigned to species (or as closely as possible) by using the *assign_taxonomy.py* script implemented in QIIME 1.9.1, with the “blast” parameter and UNITE database v7 as reference [29].

### 2.4. Statistical Analyses

Prior to the following analyses, we omitted those OTUs that belonged to the genus *Malassezia*, because this genus comprised mainly fungi from the human skin (accounting for nine OTUs and 15.2% of all reads). We also excluded four samples due to equivocal labelling and three sample that had zero OTU reads. The resulting dataset can be found in Appendix A.

For each of the three focal tree species, we calculated (i) the number of OTU reads per endophyte phylum, (ii) the summed OTU richness per endophyte phylum, (iii) the mean rarefied OTU richness per plot (rarefied to the observed minimum of 13 reads per plot) and (iv) the mean exponential of the Shannon diversity of OTUs per plot ($D=\exp\left(-\overset{n}{\underset{i}{\sum }}p_{i}lnp_{i}\right)$ with *p_i_* = proportional abundance of the *i*th OTU from a total of *n* OTUs per sample). Rarefaction curves and rank abundance curves were used to assess species-level OTU richness and evenness.

The effects of the plot-level tree neighborhood on the rarefied OTU richness were analyzed with separate linear models. The effects of the species richness and functional composition of the plot-level tree neighborhood (deciduous, evergreen or mixed) were modelled with linear models (separately for each of the three focal tree species). The effects of admixing a second tree species were modelled across all focal species. The effects of the plot-level tree neighborhood on the composition of OTUs were investigated with a distance-based redundancy analysis. Prior to the analysis, we standardized the number of OTU reads with a Hellinger transformation that reduced the impact of rare OTUs ($y_{ij}^{’}=\sqrt{\frac{y_{ij}}{y_{i+}}}$ with *y_ij_* = matrix of *i* sample rows and *j* OTU columns and *y_i+_* = row sums, [30]). Distance between samples was quantified with the Bray–Curtis distance. The significance of the effects of the identity of the focal tree species as well as the species richness and composition of the plot-level tree neighborhood on the resulting ordination of the OTU communities was assessed with a marginal test against 999 permutations [31].

Spatial distances between samples were quantified with Euclidean distance matrices (separately for Block A and Block B of the experiment). Compositional similarities between samples were quantified with the Sørensen similarity index [32]. Relationships between distance matrices and similarity matrices were analyzed with Procrustes analyses and tested against 999 permutations (separately for each of the three focal tree species).

The effects of tree species mixtures versus monocultures on the abundance of individual OTUs were investigated with multiple *t*-tests (separately for each of the three focal tree species). OTUs that occurred in less than 20 percent of all samples were omitted from the analysis. Numbers of OTU reads were log transformed prior to the analysis. Significance of the differences in log-transformed read numbers between monocultures and mixtures were determined based on Holm-adjusted *p*-values.

All statistical analyses were conducted in R [33] using the following packages: *BiodiversityR* for species accumulation curves [34], *ggplot2* for graphical representations [35] and *vegan* for rarefied species richness, redundancy and Procrustes analyses [36].

## 3. Results

In total, our analyses yielded 2,403,515 reads that could be assigned to 226 endophytic OTUs from the following phyla (Figure 1a,b): Ascomycota (122 OTUs), Basidiomycota (93 OTUs), Mortierellomycota (six OTUs), Rozellomycota (six OTUs), Mucoromycota (one OTU). The complete list of OTUs can be found in Appendix A. Beech samples had the highest number of host-specific OTUs, followed by that in spruce and oak samples, respectively (Figure 1c).

Beech, oak and spruce samples did not differ in the mean rarefied richness or Shannon exponent of OTU reads (Figure 2a,b, F_df 2, 131_ = 1.42 with *p* = 0.25 and F_df 2, 131_ = 0.67 with *p* = 0.51, respectively). Rarefaction curves indicated that the highest richness of OTUs was found for beech, followed by spruce and oak samples (Figure 2c). Rank abundance curves did not differ between the three focal tree species (Figure 2d).

On the plot-level, rarefied OTU richness was not related to the species richness of trees (Figure 3a, species richness F_df 1, 42_ = 3.61 with *p* = 0.06, F_df 1, 44_ = 0.45 with *p* = 0.51 and F_df 1, 42_ = 0.11 with *p* = 0.75 for beech, oak and spruce, respectively) or the functional composition of the tree neighborhood (Figure 3b, F_df 1, 42_ = 1.35 with *p* = 0.25, F_df 1, 44_ = 0.02 with *p* = 0.89 and F_df 1, 42_ = 0.33 with *p* = 0.57 for beech, oak and spruce, respectively). Rarefied OTU richness in two-species mixtures was not related to the identity of the admixed tree species (Figure 3c, F_df 5,24_ = 0.71 with *p* = 0.63).

The composition of the OTU community was significantly related to the identity of the focal tree species (Figure 4, marginal effect: F_df 2_ = 4.58 with *p* < 0.01) and unrelated to the functional composition and species richness of the tree neighborhood (F = 1.2, with *p* = 0.21 for the functional composition and F = 0.69 with *p* = 0.85 for tree species richness). Between samples, the similarity in OTU composition was not related to the spatial distance between sampling plots (Beech: *p* _Block A_ = 0.68, *p* _Block B_ = 0.99, Spruce: *p* _Block A_ = 0.97, *p* _Block B_ = 0.06 and Oak: *p* _Block A_ = 0.78, *p* _Block B_ = 0.61).

Holm-adjusted multiple *t*-tests revealed that five OTUs in beech samples, three OTUs in oak samples and five OTUs in spruce samples were significantly more abundant in mixed than in monospecific plots (Figure 5). None of the sampled OTUs were overabundant in monospecific plots. Three OTUs were overabundant in mixtures of more than one host tree species. Eight and two out of these 10 mixture-specific OTUs belonged to the Ascomycota phylum and the Basidiomycota phylum, respectively.

## 4. Discussion

In this study, we investigated the effects of the diversity and composition of the plot-level tree neighborhood on the composition and diversity of foliar endophytic fungi (FEF) in three focal tree species within a densely planted tree diversity experiment. Our results mostly defied our hypotheses. FEF diversity did not differ among the samples of beech, oak and spruce and was independent of the diversity and composition of the local tree community. Although FEF composition was dependent on the focal tree species, we found no effect of the functional composition or the species richness of the tree neighborhood.

Our results of a different composition but similar diversity of FEFs in beech, oak and spruce comply with the findings of leaf litter fungi in central European forests [37]. Our main findings, however, confirmed neither the published positive [15] or negative effects of tree diversity on FEF diversity [17]. Such an inconsistency in host–endophyte diversity relationships has also been reported from samples of Norway spruce in mature forests [38] and from the largest subtropical tree diversity experiment in China [18].

The observed independence of FEF diversity from tree diversity might be attributed to multiple factors. First, we do not know to what degree the sampled FEF were comprised by generalists or more specialized species [38,39]. We found thirteen OTUs that were overabundant in tree mixtures and no OTUs that were significantly restricted to tree monocultures, indicating a higher proportion of generalist species. A high proportion of generalist FEF might diminish any signal of niche- versus neutrality-based assembly processes [40]. Still, our results tentatively suggest that the three focal trees species harbored distinct FEF communities. However, we are unaware of any findings regarding the distribution of generalist versus specialist FEF in European forests.

Secondly, we did not account for the effect of the host genotype on the FEF community, albeit it has been shown that the host genotype significantly affects the FEF composition in spruce [41] and aspen trees [42].

Thirdly, the number of replicates or the amount of sampled leaf material might have been too small to capture the entirety of the FEF community, potentially omitting those FEF that are rare or highly specialized. The rarefaction curves indeed indicated that our sampling strategy might have disregarded a significant proportion of FEF species that might occur in single host individuals. Studies that used ITS markers to investigate FEF in samples of spruce leaves found a similarly low number of 2–6 OTUs per sample [41] and reported that more than 60 samples might be needed to approach complete sampling of the FEF community [43].

Fourthly, the trees in the Kreinitz experiment were planted at an initial distance of 80–100 cm. In April 2017—the beginning of the growing period— the trees have reached sizes of >5.3 ± 1.8 m (mean and standard deviation), and the branches completely overlapped (see Appendix A) so that spillover effects between neighboring individuals might have overshadowed any effects of tree species identity and diversity. However, our results showed that the composition of FEF was, at least partly, driven by selective forces from the identity of the focal tree species. Here, the absence of a relationship between the spatial distance and the compositional similarity of samples could either indicate that FEF are not distributed at random [44] or that the size of the Kreinitz experiment is too small to detect spatial patterns in FEF distribution. Similarly, J. B. Vincent et al. [45] did not observe a distance decay in endophyte similarity within 10–100 m of a tropical forest.

Lastly, we do not know to which extent the sampled FEF exhibited interspecific interactions or competition for space and resources. In pine species, for example, the competition of endophytic fungi for key metabolites can lead to the suppression or exclusion of pathogenic fungi [46]. Newcombe [47] similarly speculated that endophyte–endophyte interactions might be mainly based on competitive exclusion. However, Newcombe [47] also noted that foliar endophytes might be also very restricted in their colonization abilities, which would render any competitive interactions rather unlikely. In our experiment, none of the recorded OTUs were restricted to monospecific tree neighborhoods. On the contrary, we found that 13 OTUs were overabundant when the respective host species was growing in mixture with other tree species. Such a restriction to tree mixtures might indicate that in the Kreinitz experiment, the endophytes need a strong spatial overlap between different tree species to guarantee a spillover between distinct species. However, the experiment was already densely grown (see Appendix A), trees from neighboring plots started to touch branches and our results did not support a distance decay in endophyte similarity across the experiment. We thus speculate that endophytes should, in principle, be able to reach the monoculture plots from adjacent mixture plots. Perhaps those endophytes were unable to colonize the monoculture plots due to competitive suppression or negative interaction with rather specialized competitors. 

## 5. Conclusions

In this study, we analyzed a tree diversity experiment with the aim to assess the effects of the local tree neighborhood on the FEF community. Previous research has shown that both tree host characteristics (species identity, age, genotype and vitality) and local climatic conditions shape the FEF community. However, it is still unclear whether the local tree neighborhood additionally affects the diversity and composition of FEF. To generalize the effects of tree diversity on FEF across host conditions, tree species, forest types and abiotic gradients, more studies in controlled environments or tree diversity experiments are needed. As these factors might even interact in moderating not only the effects of tree diversity on FEF but maybe also the effects of FEF on forest functioning, the mechanisms leading to the observed distribution and the role of FEFs for forest functioning remain yet to be uncovered.

## Figures and Tables

**Figure 1 life-11-01081-f001:**
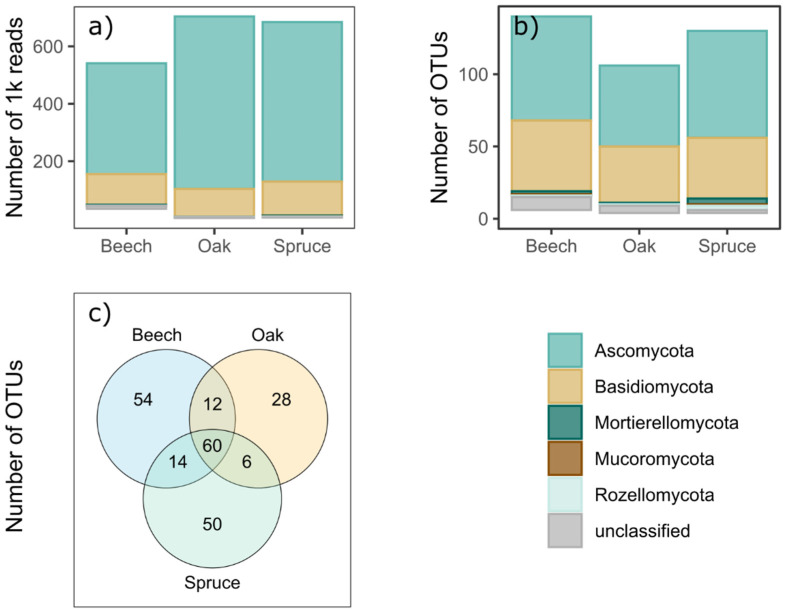
Data overview on (**a**) the number of OTU reads, (**b**) the summed OTU richness and (**c**) the number of unique or shared endophytic OTUs between the three focal tree species in the Kreinitz tree diversity experiment. Overlapping areas show the number of shared OTUs.

**Figure 2 life-11-01081-f002:**
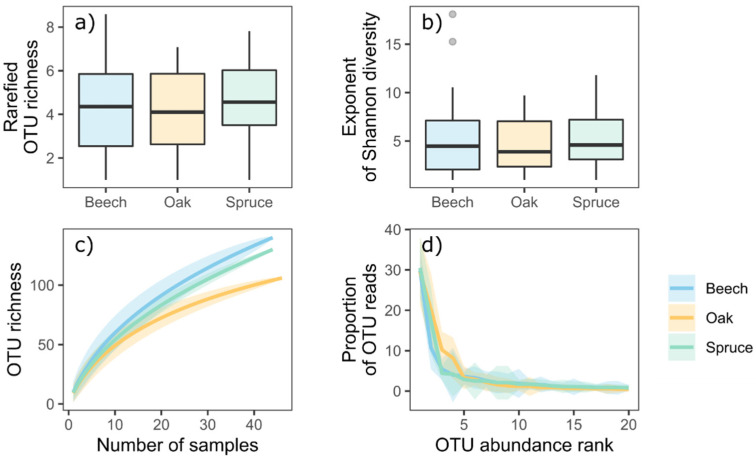
Analysis on the richness of foliar endophytic fungi in the three focal tree species in the Kreinitz tree diversity experiment. (**a**) and (**b**) show the rarefied OTU richness (tailored to the minimum of 13 reads) and the exponent of the Shannon diversity, both determined on the plot-level. (**c**) shows the rarefaction curves across all samples. (**d**) shows plot-level averaged rank abundance curves.

**Figure 3 life-11-01081-f003:**
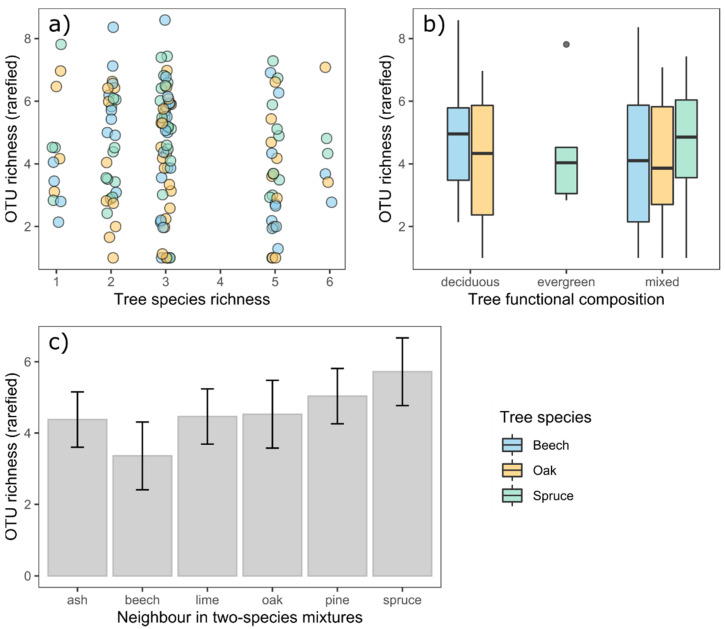
Analysis on the effects of the plot-level tree diversity on the rarefied species richness of endophytic OTUs (tailored to the minimum of 13 reads). (**a**) shows the effect of tree species richness within the three focal tree species as calculated from separate linear models, (**b**) shows the effect of the functional composition (deciduous, evergreen or mixed) and (**c**) shows the effect of the admixed tree species in two-species mixtures (±standard error).

**Figure 4 life-11-01081-f004:**
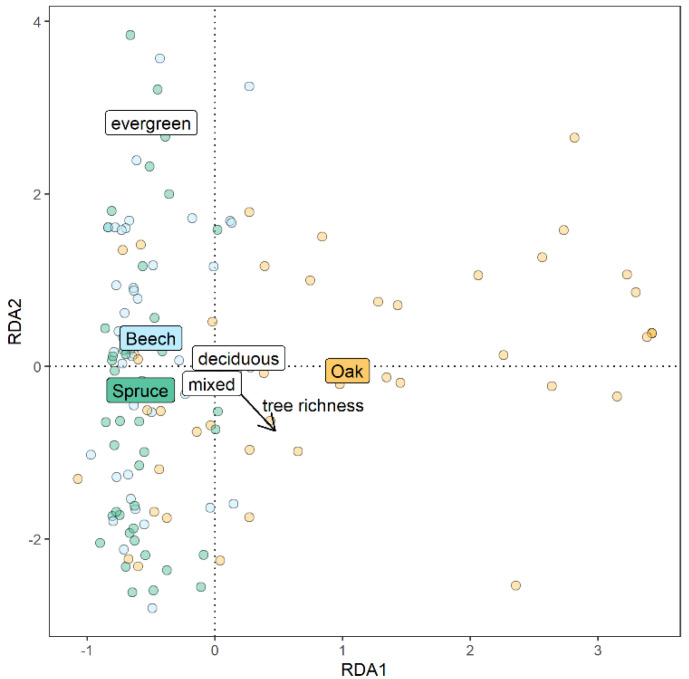
Distance-based redundancy analysis showing the effects of the identity of the focal tree species, species richness and the functional composition of the plot-level tree neighborhood (deciduous, evergreen or mixed) on the composition of endophytic OTUs (number of reads were Hellinger-transformed and standardized prior to the analysis).

**Figure 5 life-11-01081-f005:**
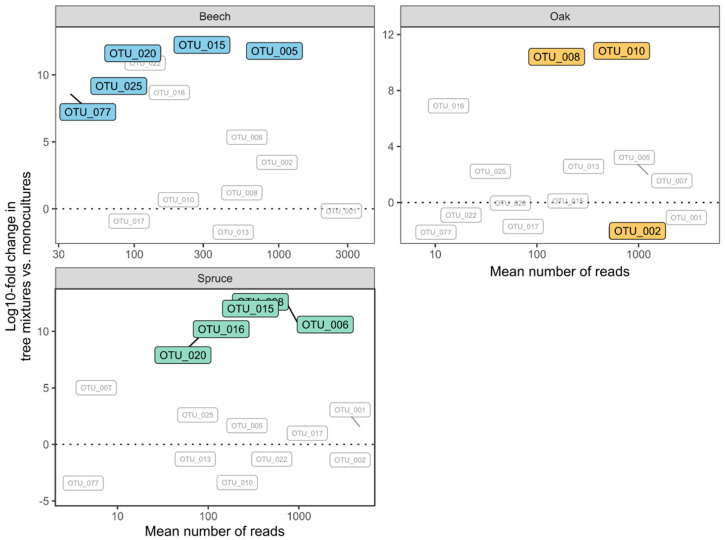
Log10 fold changes in endophytic OTUs in samples of beech, oak and spruce in mixed versus monospecific tree neighborhoods. Positive values indicate higher abundance in mixtures. Significant changes (based on multiple *t*-tests with Holm-adjusted *p*-values) are highlighted in color.

## Data Availability

All data generated or analyzed in this study are included in this published article (and its Appendix A). DNA sequences are stored at www.ebi.ac.uk/ena/browser/view/PRJEB45840.

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
