# Peer review of "Foliar Fungal Endophytes in a Tree Diversity Experiment Are Driven by the Identity but Not the Diversity of Tree Species"

_life, 2021, doi:10.3390/life11101081_

Round 1

Reviewer 1 Report

The present article foliar fungal endophytes in a densely planted tree diversity experiment are driven by the identity and composition but not by the diversity of the tree neighbourhood”  discussed the effect of surrounding  tree on the diversity of  foliar fungal endophytes.

The research topic is   quite interesting , however I suggest author for extensive revision of  article  for the better and clear presentation.

  • Title- please modify the title for clarity or modify the word identity  with  “genotype”
  • Add some details about cryptic  or conspicuous organism groups (line 39-40)
  • Reframe the sentence (48-51)
  • Line 103- remove the word “in after noon”… this paragraph deals sampling for endophyte isolation, check this paragraph
  • Statistical analysis
  • Line 141-  “Prior to the following analyses, we omitted those OTUs that had less than 20 reads”   what was the  percentage of  20 reads in total OTUs
  • “OTUs that belonged to the genus Malassezia, because this 142 genus comprised mainly fungi from the human skin (accounting for nine OTUs and 15.2 % of all reads” -Presence of Malassezia  (15.2 %)  is very surprising and it  seems your extracted DNA sample was not pure or contaminated
  • Line 191-196, statistics should be mentioned in the figure  footnote  or inside the figure that will be easy for readers  in understanding . Current form is  very lengthy and hard to understand

Reviewer 2 Report

Review for Kambach et al. “Foliar fungal endophytes in a densely planted tree diversity experiment are driven by the identity and composition but not by 3 the diversity of the tree neighborhood”, Manuscript ID: life-1371548

General comments:

I have several concerns about this study:

  • Skin-specific contaminants were introduced accounting for 15% of the data, how to insure non-skin-specific contaminants were not introduced?
  • The average number of OTUs is worryingly low with ca. 4 OTUs per sample (Figure 2) and many samples having just 1 or 2 OTUs (Supplement data1). I am aware of an endophyte study using the same primers obtaining ca. 30 OTUs per sample [1]. I don’t have time to look up more references, but the authors should see how well their findings fit similar endophyte studies.

[1] Unterseher et al. 2018. Mycobiomes of sympatric Amorphophallus albispathus (Araceae) and Camellia sinensis (Theaceae) – a case study reveals clear tissue preferences and differences in diversity and composition. Mycological Progress.

  • Given this low richness per sample the applied analysis such as RDA ordination and log2foldchange is inappropriate.
  • I believe the results obtained from modelling the log2fold change is unreliable (see me elaborated comments on this issue below).

I do not find the results as they are presented here reliable and will therefore not evaluate the discussion at this point.

Specific comments:

L 17. the word “trigger” seems misused here and the “up to” is missing a “from” to form a full sentence. I would rephrase to something like: “Symbiotic foliar fungal endophytes can have beneficial effects on host trees, and might alleviate climate-induced stress” (from your introduction L. 49 I understand that it is not yet shown)

L. 22. Add hyphens for redundancy-, Procustes- and differential abundance analysis.

L. 51 I suggest rephrasing to something like: “To predict the effects of tree diversity on forest diversity and functioning, its effect on the FEF community must also be understood”

L. 53-54 this sentence is very vague consider highlighting more specific knowledge from the reference no. 6 (review by Qie Jia et al 2020).

L. 53 to 68 I want to point the authors to the work of Eusemann et al 2016 [2] and Würth et al 2019 [3] their findings seem relevant in this context, having analyzed FEF diversity and composition (including epiphytes) of Picea glauca stands with regards to host tree genetics, - age (incl needle age) and – size, forest core- or edge stands and other environmental parameters.

 [2] Eusemann et al 2016. Habitat conditions and phenological tree traits overrule the influence of tree genotype in the needle mycobiome – Picea glauca system at an arctic treeline ecotone. New Phytologist

[3] Wurth et al 2019. The needle mycobiome of Picea glauca – A dynamic system reflecting surrounding environment and tree phenological traits. Fungal Ecology.

L. 74 specify that the study by Hantsch et al 2014 was based on microscopy.

L. 75 I suggest rephrasing to something like: “in this study we a molecular marker (ITS) to identify the FEF community”

L. 100 note typo wigs / twigs.

L. 102 “stored on ice”, I think it means samples were kept cool at ca. 4 °C, or were they frozen? please specify.

L. 104 “bidest. water” I think it is more commonly referred to as “double-distilled water” or simply ddH2O

L. 103 to 108, does a reference exist for this protocol, if yes, please add.

L. 111 can you add ca. how many grams the dried material (the 32 discs/needles) weighted?

L. 124 Primers ITS1F and ITS4 produce a fragment of ca. 800 bp (including ITS1, 5.8S, ITS2). It was sequenced using 2x250bp, which is not long enough to obtain an overlap, thus only forward reads of the ITS1 region (ITS1) could be used. This is not an optimal approach or usage of paired-end sequencing but acceptable.

Paragraph 2.3. This description of the bioinformatic is ultra-short and does not allow for a replication of the procedure since no information about quality parameters etc. is given. I advise to either elaborate the entire paragraph or add a detailed supplement file with the bioinformatic pipeline. Also, it is not mentioned with what classifier the OTUs were assigned only that UNITE was used as database.

L.141 how many reads were obtained per sample? Were the samples rarified prior to this exclusion of OTUs < 20 reads, please specify.

L. 142 the presence of Malassezia sp. point to an unfortunate contamination of the data, representing 15% of the reads that is a lot, how can you be sure that same skin didn’t carry non-skin specific contaminants?

L.146 “OTU” seems to be misplaced, it should be “… the summed OTU richness and number of reads per endophyte phylum…”.

L. 147 I don’t understand how to relate the information in the brackets “(for the minimum of 13 reads per sample)” to OTU richness. And please refer to “richness” as “OTU richness” for clarity (the paragraph is not easy to read consider improving overall readability).

L. 154 change the slash to a comma separating deciduous and evergreen, as they are not considered the same thing.

L. 155 should it be understood that the addition of a third, fourth and fifth tree was not analyzed? (that is how I read it).

L. 160 please add the squared brackets for the reference 27.

L. 173 I have concerns about the use of DESeq2 to model log2fold changes, although widely applied in the field of molecular microbial ecology the method is often not appropriate for this kind of data. I have personally invested a lot of time into this method, as I wanted to apply it myself. I have had personal contact with developers Michael Love (Biocondiuctor) and Susan Holmes (phyloseq) and they both said that microbiome data (in general) is too sparse (too many zeros) which is messing up the model fitting, and if using the model you should apply very strong filtering of your OTU abundance. And indeed, when evaluating my own findings from the DESeq2 output I found that they were not true! Only when applying a filter of presence in min. 50% of my samples did I obtain reliable results. If it is any help I would be happy to share my evaluation (some powerpoint slides illustrating the issues and potential solutions) with you (in that case contact the editor).

An additional comment is that mono-culture stands vs. mix-cultures is, as I understand the plot design, an unbalanced comparison, there are many more mixed plots than mono plots, right? In that case you can have an OTU is perhaps 3 or 4 of the mixed plots and not in the mono-culture plot, but that could very likely be by chance.

The fastest way to visualize whether you have problem is to plot the so-called MA-plots, like this:

# Apply the Deseq algorithm
dds<- DESeq(deseq_c,fitType = "local") # First check your model fit, parametric, local or mean (I refer you to the deseq tutorial for that)

# apply schrinking (when necessary)
resLFC <- lfcShrink(dds, coef="site_GR_vs_BH", type="apeglm")

# plot MA-plots
DESeq2::plotMA(resLFC, cex= 1)
DESeq2::plotMA(res,cex= 1) # without shrinking

Unfortunately based on you figure 5, I can tell that you do have problems with the model fitting in all three tree species.

My suggestion is to either apply a strong filter or to abandon the DESeq2 modelling and instead calculate the log2fold change manually and make appropriate choices for statistics (e.g. dealing with multiple testing and such), the latter would be my personal choice for an approach.

Now I had a closer look into your data table 1 and you have a very very high frequency of zeros, and as you show in your figure 2 your average number of OTU per sample is around 4 OTUs, it that case I consider ordinations such as that in figure 4 to be inappropriate. How is it logical meaningful to compare sample community similarity when the communities have such few members and, in several cases, only consists of 1 or 2 OTUs? It is not.

Figure 1 correct abbreviation for thousands, tsd is German. K would be the English, but it is an unusual usage, maybe just write thousands.

Figure 3a I don’t see any justification to add regression lines here.

I do not find the results as they are presented here reliable and will therefore not evaluate the discussion at this point.

Reviewer 3 Report

The manuscript wrote by Stephan Kambach et al showed an interesting work about foliar fungal endophytes in a densely planted tree diversity experiment. Although the experiments are not very complex, this work focuses on page fungi, which are often overlooked, is innovative and the authors provide a thorough analysis and discussion of the sequencing data. The English is also very good and very comfortable to read. There are only a few small areas that need to be revised. After this, it is possible to publish what I think

  1. The title is so long that I suggest distilling the most important parts of the paper.
  2. There are too many keywords, I suggest cutting down to six.
  3. p should be italic in the MS
  4. Line216 There are two “in” in this sentence and please remove one.
  5. Line 217 I think that “between” should change to “among”
  6. I suggest that the author add a clearly conclusion in the end of the MS.

Reviewer 4 Report

Dear Authors,

The article titled " Foliar fungal endophytes in a densely planted tree diversity experiment are driven by the identity and composition but not by the diversity of the tree neighborhood" provided a lot of useful information about how the tree neighborhood effects on the diversity and composition of the foliar endophytic fungi in beech, oak and spruce. The authors are experienced in this field, which proves their previous studies focused on tee and fungal diversity. The article is interesting and may open new directions in biodiversity contribution to forest ecosystem functioning and the provision of ecosystem services. The experiment is well planned and implemented. The discussion of the results is correct and clearly written. In my view, the title should be modified to be more attractive to readers. I am strongly suggested to publish this article as it is in the life journal.

Regards

Ahmed Khalil

Round 2

Reviewer 1 Report

Paper can be accepted in the present form

Author Response

We would like to thank the reviewer for his first comments and the positive evaluation now.

Reviewer 2 Report

L110 bidest. water change to ddH2O to be consistent.

L. 188 remove reference to DeSeq2 as it is no longer used.

L.289 the phrasing "it is clear..." makes it seem like the following statement is based on your data, I suggest rephrasing or keeping the focus in the conclusion exclusively to your findings.

Author Response

We would like to thank the reviewer for his first comments and adapted the  three suggested minor changes.

L110 bidest. water change to ddH2O to be consistent.

Response: We changed bidest. water to ddH20.

L. 188 remove reference to DeSeq2 as it is no longer used.

Response: We removed the reference to the Deseq2-package

L.289 the phrasing "it is clear..." makes it seem like the following statement is based on your data, I suggest rephrasing or keeping the focus in the conclusion exclusively to your findings.

Response: We rephrased this sentence to: "Previous research has shown that ..." to make it clear that these are not our results but previous findings, which are all stated in the introduction.